# Rechargeable Concrete Battery

**Emma Qingnan Zhang *** and Luping Tang

Division of Building Technology, Department of Architecture and Civil Engineering, Chalmers University of Technology, 41296 Gothenburg, Sweden; tang.luping@chalmers.se
* Correspondence: emma.zhang@chalmers.se

**Abstract:** A rechargeable cement-based battery was developed, with an average energy density of 7 Wh/m$^2$ (or 0.8 Wh/L) during six charge/discharge cycles. Iron (Fe) and zinc (Zn) were selected as anodes, and nickel-based (Ni) oxides as cathodes. The conductivity of cement-based electrolytes was modified by adding short carbon fibers (CF). The cement-based electrodes were produced by two methods: powder-mixing and metal-coating. Different combinations of cells were tested. The results showed that the best performance of the rechargeable battery was the Ni–Fe battery, produced by the metal-coating method.

**Keywords:** concrete battery; rechargeable; carbon fiber; conductive cementitious material; electroplating; energy storage; functional composites

## 1. Introduction

Advanced building materials of the future are being envisioned to provide multifunctional smart features such as self-powering [1] and self-sensing [2,3] for structural health monitoring applications. Moreover, future building materials could also take on additional functions such as renewable energy sources [4], which will collect and store renewable energy such as solar and wind energy. The concept of using structures and buildings as energy source and storage could be revolutionary, because it offers an alternative solution to solve the energy crisis by providing a large amount of energy storage. Due to the large volumes of structures, the capacity of energy storage can be high, even if the energy per unit volume is not high.

Many researchers have devoted much effort to developing electrical properties of concrete to meet future requirements, which is the most used construction materials in the world. By adding electrically conductive components, stable and good electrical conductivity of concrete can be achieved [5–7], and concrete can obtain functions such as supercapacitors [8] and batteries [9].

Previous works on cement-based batteries have been focused on non-rechargeable types, defined by the electrochemical processes, while if categorized by the battery arrangement, there have been mainly two types: dispersed-particle type and electrode-probe type. The first proof of concept of cement-based batteries demonstrated by Meng and Chung [9] was a dispersed-particle type of battery, which means the active metal particles were mixed and imbedded in a cement paste matrix. In their work, the active metals were manganese dioxide ($MnO_2$) particles as cathode material and zinc (Zn) particle as anode material. Carbon black was added as a functional filler in the cement paste matrix to improve the conductivity. The output of this battery was very low, with an initial open-circuit potential up to 0.72 V with a peak current of 120 μA, and the initial power output was 1.42 μW/cm$^2$.

Researchers have been continuously making progress to develop concrete batteries. Using the same electrode materials as in Meng and Chung's work, multi-scale carbon-based fillers were added to further improve the conductivity and stability of the concrete matrix. The maximum current output was achieved at 250 μA (35.21 μA/cm$^2$) over 24 h [10]. Holmes et al. [11] examined batteries with similar constituents and found that the Epsom

salt-cured batteries had decreased output and a shorter lifespan, which indicated that maintaining sufficient water content is essential for the performance of the concrete battery. Other combinations of electrode materials, conductive fillers, and different water-cement ratios (w/c) were investigated as well. Byrne et al. [12] found that the optimal electrode materials were magnesium anode and copper cathode in their study, and they concluded that increasing the w/c ratio could slightly increase the current output but had no effect on prolonging the lifespan. Their battery had an initial peak current of 4.37 mA (over a 10 $\Omega$ resistor) and a duration of 12 h (measured when current was great than 0.59 mA).

Efforts have also been made to develop the immersed electrode-probe type of batteries, in which the hardened cement paste or mortar is the electrolyte. Burstein and Speckert [13] developed an alkaline aluminum/water battery, which employed the reduction of water to hydrogen on steel as a cathode reaction. This battery could provide a very low current density, and the highest voltage recorded was 0.6 V. Ouellette and Todd [14] developed a cement seawater battery energy harvester with magnesium (Mg) and carbon probe electrodes for low-power generation applications for monitoring marine infrastructure subject to corrosion.

Research of concrete batteries is still rare, and the performance of the batteries is very poor. Moreover, there have not been any investigations on rechargeable concrete batteries. Therefore, in our work, we have chosen to challenge this topic with the aim of demonstrating the feasibility of developing rechargeable concrete batteries with improved power output and energy capacity.

## 2. Materials and Methods

### 2.1. Battery Design

In order to optimize electrochemical cells in a highly alkaline concrete environment, we identified the following metals that are suitable for rechargeable concrete batteries. The alternatives for anode materials are iron (Fe) and zinc (Zn), both of which undergo reduction during charging and oxidation during discharging. Their half-cell reactions are shown in Table 1, where $E_{red}°$ stands for standard reduction potentials. It should be noted that zinc can also turn to non-rechargeable zinc oxide during discharge. The possible choices for cathodes are, however, very limited. Although manganese hydroxides have the possibility for recharging, they can easily turn to non-rechargeable manganese oxides. On the other hand, manganese hydroxides are not commercially available. Therefore, the only choice is nickel, which can turn to nickel oxide hydroxide and nickel oxide under alkaline environments by an oxidation reaction during charge and reduction during discharge. Their half-cell reactions are shown in Table 2, where $E_{red}°$ stands for standard reduction potentials.

**Table 1.** Possible anode materials and their half-cell reactions.

| Metal | Half-Cell Reaction | $E_{\textbf{red}}°$ **(V)** |
|---|---|---|
| Iron | $Fe(OH)_{2(s)} + 2e^- \rightarrow Fe_{(s)} + 2OH^-_{(aq)}$ | −0.89 |
| Zinc | $Zn(OH)_4{}^{2-}{}_{(aq)} + 2e^- \rightarrow Zn_{(s)} + 4OH^-_{(aq)}$ | −1.20 |

Because the aim of the study is to develop rechargeable batteries, we have therefore excluded other options for non-rechargeable batteries. In this work, we focused on iron, zinc, and nickel-based oxides and tested different combinations of them.

After selecting the cell reactions, the next step is to design the mixture for conductive mortar in the electrode layers. The battery arrangements have been mainly two types: layered structures [9] and immersed [15], as shown in Figure 1. In this work, we used the layered structure of the battery, which consists of an anode layer, a separator (or electrolyte) layer, and a cathode layer. Besides the conventional way of mixing metal powder particles with cement paste, we developed another approach to make the electrodes, namely electroplating metals onto carbon fiber (CF) meshes and then casting the metal-

coated CF meshes in the conductive mortar. In the following text, we refer to these two methods as the powder-mixing method and the metal-coating method.

**Table 2.** Possible cathode materials and their half-cell reactions.

| Metal | Half-Cell Reaction | $E_{red}^{\circ}$ (V) |
|---|---|---|
| Nickel oxyhydroxide | $NiOOH + H_2O + e^- \rightarrow Ni(OH)_2 + OH^-$ | +0.52 [16] |

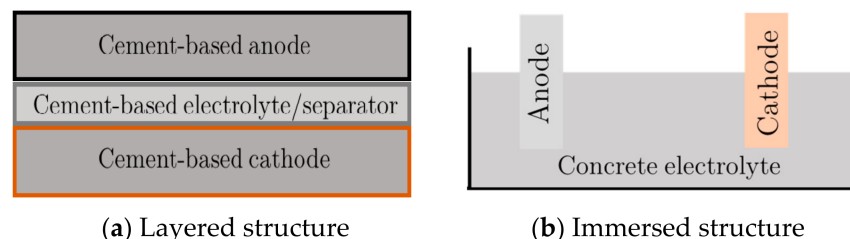

(**a**) Layered structure    (**b**) Immersed structure

**Figure 1.** Two common cell arrangements of the concrete batteries: (**a**) the layered structure and (**b**) the immersed structure.

The powder-mixing method means that the active electrode materials (metal particles) were added to the cement mixture, and the electrode layer was formed after the hydration of the cement paste, as shown in Figure 2. The metal-coating method refers to the active metals being electro-plated onto the CF meshes and the metal-coated CF meshes being cast inside a layer of conductive cement-based mortar to form the anode or cathode layer, as shown in Figure 3. The CF meshes were PAN-based and commercially available. The spacing of the meshes was 10 mm by 8 mm with a thickness of 1 mm. The fiber bundle in the longitudinal warp direction was 12K (12,000 filaments per fiber tow) and in the vertical weft direction was 6K. The detailed information of the compositions for the polymer matrix in the CF meshes is protected. However, the proportion of the polymer matrix was 15% by weight.

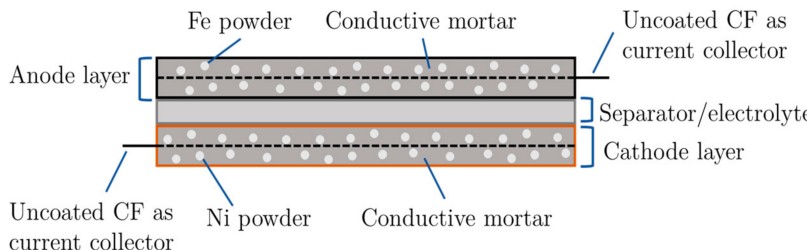

**Figure 2.** Schematic illustration of the design of the powder-mixing battery.

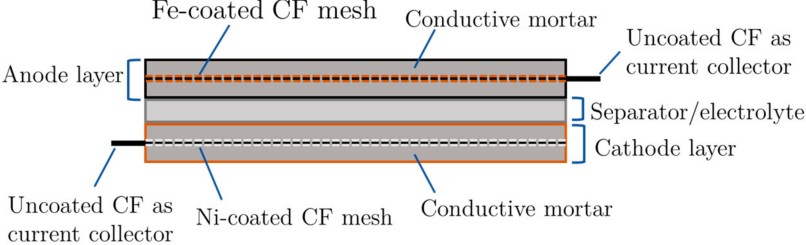

**Figure 3.** Schematic illustration of the design of the metal-coating battery.

For the mixture of the cement-based electrolyte separator, we added ion exchange resin to the cement mortar to increase the ionic conductivity, especially hydroxide ions,

while also providing high electrical resistivity to the cement-based electrolyte separator. The ion exchange resin is a commercially available product, IRA-402(OH). An example of the mixture design of the cement-based electrolyte separator is presented in Table 3, and the composition of the alkaline solution is shown in Table 4.

**Table 3.** Mix proportion of the cement-based electrolyte separator.

| Cement (g) | Fine Sand (g) | Alkaline Solution (g) | Ion Exchange Resin (g) | Total Volume (cm³) |
|---|---|---|---|---|
| 22 | 15 | 12 | 8 | 35 |

**Table 4.** Compositions of the alkaline solution.

| KOH (85.6%) (g) | LiOH·H$_2$O (>98%) (g) | Deionized Water (g) |
|---|---|---|
| 245.3 | 12.6 | 756.3 |

### 2.2. Mixture of Conductive Mortar

Carbon fibers (CFs) are electrically conductive. When CFs are used as additional filler in cement-based materials, they can increase electrical conductivity, decrease drying shrinkage, and increase flexural toughness. In this study, we chose short carbon fibers as functional fillers for the cement mixture to form the conductive cementitious anode or cathode layers for concrete batteries. The reasons that we decided to exclude the use of other carbon-based fillers, such as carbon black or carbon nanotubes, were firstly their high price, which will be a great obstacle for large-scale production in the future, and secondly the difficulty in dispersion and subsequently the significant adverse effect on the workability [17].

A low fiber volume fraction is preferred; as the material cost increases, the workability decreases [18]. Hence, a good mixture for conductive cement-based mortar needs to combine both good workability and conductivity. Therefore, different proportions of the mixtures were tested to find out the optimal mixture, as shown in Table 5. Moreover, for the plain reference sample, we tested samples containing different amounts of CF and methylcellulose as the CF dispenser. The water-cement ratio (w/c) of all the samples except for one, which had a high w/c of 1.2, was 0.6. After 7 days of curing in water, the resistivity of the conductive cement-based mortar was measured.

**Table 5.** Proportions for the cement-based conductive mortar and their resistivity.

| | Ref11 | Test 4 | Test 5 | Test 6 | Test 7 | Test 8 | Test 9 | Test 10 | Test 13 | Test 14 |
|---|---|---|---|---|---|---|---|---|---|---|
| Cement * (g) | 40 | 40 | 40 | 40 | 40 | 40 | 40 | 40 | 40 | 40 |
| Sand (0–1 mm) (g) | 80 | 80 | 80 | 80 | 80 | 80 | 80 | 80 | 80 | 80 |
| CF (3 mm) (g) | 0 | 0.2 | 0.2 | 0.1 | 0.2 | 2.3 | 1.1 | 0.8 | 0.6 | 0.4 |
| Deionized water (g) | 24 | 24 | 24 | 24 | 24 | 48 | 24 | 24 | 24 | 24 |
| Methylcellulose (g) | 0 | 0.25 | 0.12 | 0.07 | 0.10 | 1.38 | 0.39 | 0.24 | 0.19 | 0.19 |
| Superplasticizer ** (g) | 0 | 0.9 | 0.4 | 0.2 | 0.5 | 1.1 | 2.6 | 1.0 | 0.7 | 0.6 |
| Water-cement ratio | 0.6 | 0.6 | 0.6 | 0.6 | 0.6 | 1.2 | 0.6 | 0.6 | 0.6 | 0.6 |
| CF volume (%) | 0% | 0.2% | 0.2% | 0.1% | 0.2% | 1.4% | 0.9% | 0.7% | 0.5% | 0.3% |
| Resistivity (Ω·m) | 10.53 | 5.52 | 7.11 | 10.03 | 8.27 | 0.35 | 0.91 | 2.11 | 2.21 | 4.14 |

* Swedish Byggcement, CEM II/A-LL 42.5 R according to EN 197-1; ** CHRYSO® Premia 205.

### 2.3. Manufacture of Powder-Mixing Electrodes

The mix proportions of the powder-mixing type of electrodes are presented in Table 6 for cathodes and Table 7 for anodes. To produce functional anodes, high alkaline solution is needed in the mixture, the composition of which is given in Table 4. The method of direct powder-mixing is easy and straightforward. However, the major drawback of this method

is the health concern regarding the procedure of mixing the Ni-containing powders with cement as well as the poor electrical performance that we later found.

**Table 6.** Mix proportions of the "powder-mixing" cathodes.

| Cathode Mix Number | Ni-1 | Ni-2 | Ni-3 | Ni-4 |
|---|---|---|---|---|
| Cement 42.5R (g) | 15 | 10 | 20 | 6 |
| Deionized water (g) | 10.8 | 11.0 | 10.0 | 10.0 |
| CF (3 mm) (g) | 0.8 | 0.8 | 0.8 | - |
| CF powder (0.2 mm) (g) | 0.35 | 0.35 | 0.35 | 1.80 |
| w/c ratio | 0.79 | 1.20 | 0.55 | 1.76 |
| CF volume (%) | 2.7% | 2.7% | 2.8% | 6.2% |
| $Ni(OH)_2$ powder (g) | 22.5 | 30 | 15 | 10 |
| Metal volume (%) | 24% | 31% | 16% | 15% |
| Superplasticizer (g) | 1.5 | 1.5 | 1.5 | 0.8 |

**Table 7.** Mix proportions of the "powder-mixing" anodes.

| Anode Mix Number | Fe-1 | Fe-2 | Fe-3 | Fe-4 | Fe-5 |
|---|---|---|---|---|---|
| Cement (g) | 12 | 14 | 6 | 6 | 6 |
| Alkaline solution (g) | 15.8 | 14.0 | 6.8 | 10.1 | 10.2 |
| CF (3 mm) (g) | - | 0.8 | - | - | 0.8 |
| CF powder (0.2 mm) (g) | - | 0.35 | 0.60 | 1.60 | 1.60 |
| w/c ratio | 1.12 | 0.87 | 0.95 | 1.39 | 1.42 |
| CF volume (%) | 0% | 2.8% | 3.3% | 6.7% | 9.5% |
| Fe powder (g) | 60 | 35 | 12 | 12 | 12 |
| Metal volume (%) | 29% | 20% | 15% | 12% | 12% |
| Superplasticizer (g) | 1.4 | 1.7 | 0.5 | 0.6 | 0.7 |

*2.4. Manufacture of Metal-Coating Electrodes*

The method of electroplating is a well-established technique that can avoid the health risk caused by dust production during direct mixing. To electroplate metal onto carbon fibers (CFs), the CF mesh works as the cathode and is connected to the negative terminal, and the metals are connected to the positive terminal as the anode. During electroplating, the metal ions of the anode will dissolve and transfer onto the surface of the cathode, which was CF mesh in this work. The conditions for electroplating are listed in Table 8. The choice of the current and the duration of electroplating was based on several try-outs, until the meshes were fully covered by the coated metals. The designed surface area of the metal-coated CF mesh was 90 mm × 90 mm. However, during the preparation, the actual coated area was larger than designed to ensure the whole surface was fully covered by metal coating. The extra surface of the meshes were cut off before casting them inside the conductive mortar electrode layers.

**Table 8.** Details of electroplating metals on carbon fiber.

| Cathode | Anode | Bath Solution | Current and Duration |
|---|---|---|---|
| CF mesh | Ni plate | 250 g/L $NiSO_4 \cdot 7H_2O$<br>20 g/L $NiCl_2 \cdot 6H_2O$<br>25 g/L $H_3BO_3$ | 1.0A for 4 h |
| CF mesh | Zn plate | 74 g Zn granulate<br>1 L acetic acid<br>20 g NaCl<br>15 g sugar | 1.0A for 4 h |
| CF mesh | Fe plate | 180 g/L $FeSO_4 \cdot 7H_2O$ | 1.2A for 6 h |

After electroplating, we were able to produce three types of CF meshes, which were iron-coated mesh (Fe–CF) and zinc-coated mesh (Zn–CF) used for the anode, as well as nickel-coated CF mesh (Ni–CF) used for the cathode. Table 9 presents the weight change of the meshes before and after the electroplating and their calculated thickness of coating. The theoretical thickness of coating was based on the weight changes and the total charges passed through the system. Iron-coated CF mesh had a thicker coating to ensure a stable performance of the anode, which we discovered from experimental results before. Figure 4 shows the images of the metal-coated CF meshes and an electrode with metal-coated CF cast inside the cement-based electrolyte. In Figure 4a,c, Ni–CF mesh and Fe–CF mesh had obvious color changes. However, for Zn–CF mesh in Figure 4b, the color was somewhat dull and not as shiny as the nickel coating. As shown in Figure 4d, after demolding from the white plastic foam, the dimension of the coated-metal electrodes was 100 mm × 100 mm × 4 mm, while the dimension of the metal-coated CF mesh was 90 mm × 90 mm × 1 mm. The electrodes were then cured in sealed plastic bags for 7 days before they were ready for the battery testing.

**Table 9.** Weight gained of metal-coated carbon fiber meshes.

|  | Ni–CF Mesh | Zn–CF Mesh | Fe–CF Mesh |
|---|---|---|---|
| Initial weight (g) | 2.433 | 3.577 | 2.433 |
| Final weight (g) | 4.955 | 6.311 | 8.714 |
| Gained weight (g) | 2.522 | 2.734 | 6.281 |
| Theoretical thickness of coating (μm) | 23 | 30 | 60 |

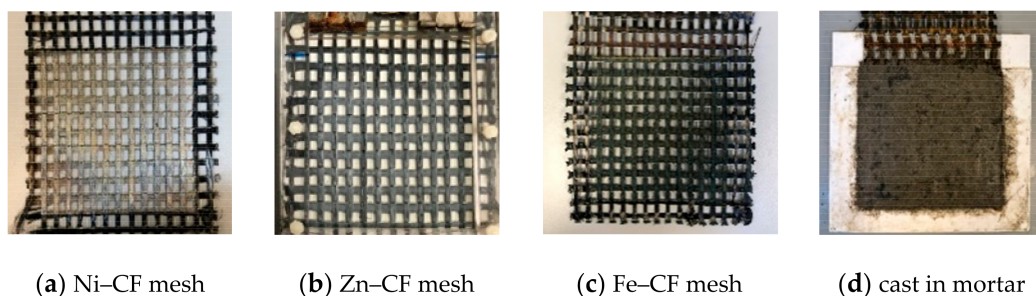

(**a**) Ni–CF mesh     (**b**) Zn–CF mesh     (**c**) Fe–CF mesh     (**d**) cast in mortar

**Figure 4.** Images of metal-coated CF meshes (**a**–**c**) and an example of coated-metal CF mesh cast inside the cement-based electrolyte before demolding from the white plastic foam (**d**). The mixture of mortar is the same for the powder-mixing mortar.

## 3. Experiments

### 3.1. Measurement of Resistivity of Conductive Mortar

The resistivity was measured using the uniaxial method, in which the paste/mortar samples of disc shape with diameter 62.5 mm and thickness 22 mm was placed between two parallel metal plates with moist sponge contacts at the interface to ensure a proper electrical connection. An LCR-meter (Keysight U7233C) was used for the measurement of resistance at a frequency of 1 kHz.

### 3.2. Measurement of the Cells and Concrete Battery

The cells were charged by a commercial charger for rechargeable Ni–Cd batteries. The charging periods were about 12–16 h, and the discharging periods were about 8–12 h (except one cycle was 22 h). The cells and concrete battery went through several charge/discharge cycles in a no-load condition first and then were tested with a resistor. The potentials of the anode and the cathode were monitored against a standard saturated calomel electrode (SCE).

It is worth emphasizing that the cells were measured with two separate electrode layers in alkaline solution (liquid electrolyte), while the structural battery was constructed

as an anode layer, a separator layer, and a cathode layer, where the electrolyte is the cement-based separator.

## 4. Results

### 4.1. Resistivity of Conductive Cement-Based Mortar

Figure 5 presents the resistivity of the conductive cement-based mortar against the CF volume. Compared to the reference sample, the resistivity was decreased as the CF volume increased as a general trend. Tests 8, 9, and 10 were excluded from further development because of their poor workability, although their resistivity seemed to be better. Particularly for Test 10, it needed to double the water content in order to be flowable, and the strength of such a mixture would be too low to use. Tests 4, 5, and 7 had the same mixture; only the amount of CF dispenser was different. As the results indicate, methylcellulose as a CF dispenser did improve the resistivity to a certain extent. Therefore, the optimal amount for CF dispenser should be around 0.8% of the water content. As a conclusion, based on the tested 10 mixtures, Test 13 showed the best combination of good workability and low resistivity so that this mixture was chosen for the conductive cement-based mortar for the anode and cathode layer in the concrete battery.

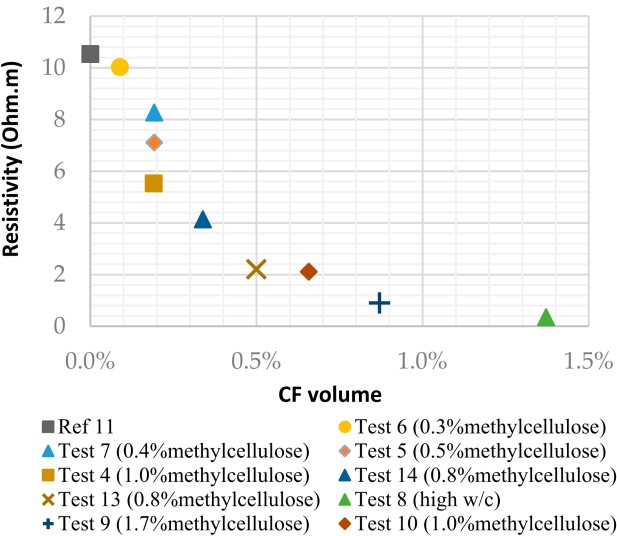

**Figure 5.** Resistivity of the cement-based solid electrolyte using different mixtures.

### 4.2. Electrical Performance of Powder-Mixing Electrodes

Table 10 gives the test arrangement of the blend-metal electrodes; the active metals were $Ni(OH)_2$ for the anodes and Fe powder for the cathode. The electrical performances of these four cells of powder-mixing electrodes were measured constantly over the charge and discharge cycles. Figure 6 illustrates one discharge cycle of these four cells. As shown in Figure 6a, the current dropped quickly after about 30 to 90 min. Among these four cells, Cell 1 performed slightly better than the others, as its current could hold at a higher level for about 90 min before it went below 1 mA. The electrical capacity of Cell 1 was 5.6 mAh during discharge through a 500 $\Omega$ resistor, which is equal to an energy density of 0.54 $Wh/m^2$ (or 0.06 Wh/L). As seen in Figure 6b,c, for Cell 1 we were able to identify that it was the cathode side that could not hold the potential at a stable level. In general, the performance of powder-mixing electrodes was not satisfactory for the potential use as a concrete battery.

**Table 10.** Cell arrangement of powder-mixing electrodes.

|  | Cell 1 | Cell 2 | Cell 3 | Cell 4 |
|---|---|---|---|---|
| Anode | Ni-1 | Ni-2 | Ni-3 | Ni-4 |
| Cathode | Fe-3 | Fe-4 | Fe-5 | Pure Fe plate |

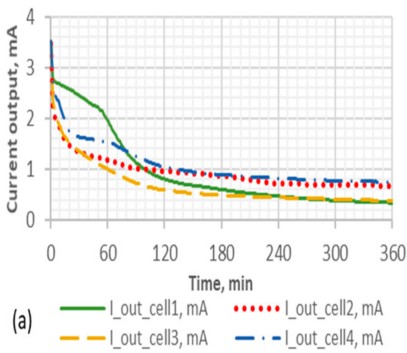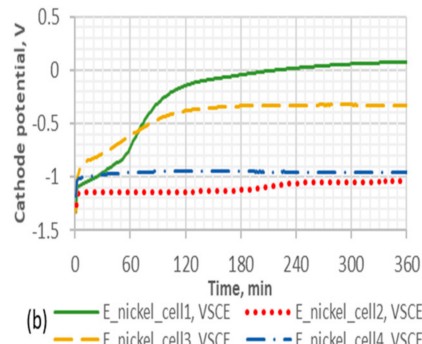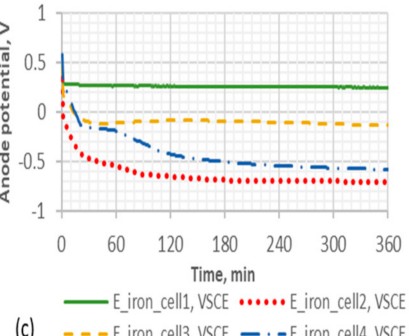

**Figure 6.** Current and potential profiles of the powder-mixing electrodes through a 500 Ω resistor for each cell. The aarrangements of the cells are listed in Table 10. Subfigure (**a**) represents the current profile, (**b**) for the cathode profile and (**c**) for the anode profile.

### *4.3. Electrical Performance of Metal-Coating Electrodes*

The Ni-CF anode was coupled with three different types of cathodes, as shown in Table 11. As the results indicated, the performance of Fe-CF anode was better than that of the pure iron plate, while the Zn–CF anode was not stable to maintain the potential during discharge.

**Table 11.** Cell arrangement of metal-coating electrodes.

|  | Cell A | Cell B | Cell C |
|---|---|---|---|
| Anode | Mortar with Ni–CF mesh | Mortar with Ni–CF mesh | Mortar with Ni–CF mesh |
| Cathode | Iron plate | Mortar with Fe–CF mesh | Mortar with Zn–CF mesh |

### 4.3.1. Ni–CF Anode vs. Iron Plate Cathode

The cement-based anode with Ni-coated CF mesh (90 mm × 90 mm × 4 mm) was tested against a pure iron plate (90 mm × 90 mm × 1 mm). The resistor under discharge was 330 Ω. The measured discharge period was 8 h, where the current was greater than 1 mA. Figure 7 illustrates the electrical performance of the Ni–CF cement anode over the 8-h discharge period. The electrical capacity of discharge cycle one (Figure 7a) was 22 mAh, and for cycle two (Figure 7b) it was 20 mAh. As compared with powder-mixing electrodes, the performance of the metal-coating anode (average electrical capacity of 21 mAh) was much better than that of the powder-mixing anode, which was only 5.6 mAh in the best case.

### 4.3.2. Ni–CF Anode vs. Fe–CF Cathode

The cement-based anode with Ni–CF mesh was tested against the Fe–CF cathode, the dimensions of both of which were 90 mm × 90 mm × 4 mm. Figure 8 presents the potential and current profile of the Ni–Fe cell during five charge/discharge cycles. Except for cycle one, the other four discharge periods were 6 h, and the average electrical capacity was 43 mAh.

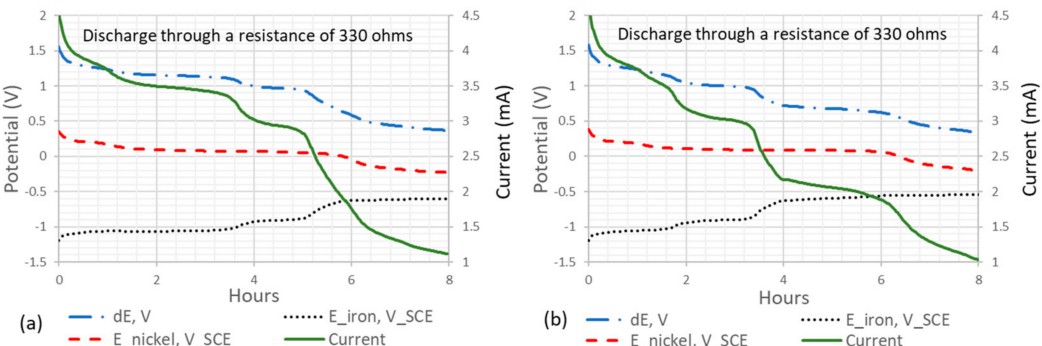

**Figure 7.** Potential and current profiles of the metal-coating Ni-CF cement-based anode against a pure iron plate cathode over two 8-h discharge cycles. Measurements include the potential difference between the anode and cathode (dE), the potential of the iron plate as the cathode (E_iron, V$_{SCE}$), the potential of the Ni-CF anode (E_nickel, V$_{SCE}$), and the current with a 330 Ω resistor. Subfigure (**a**) represents discharge cycle one and (**b**) discharge cycle two.

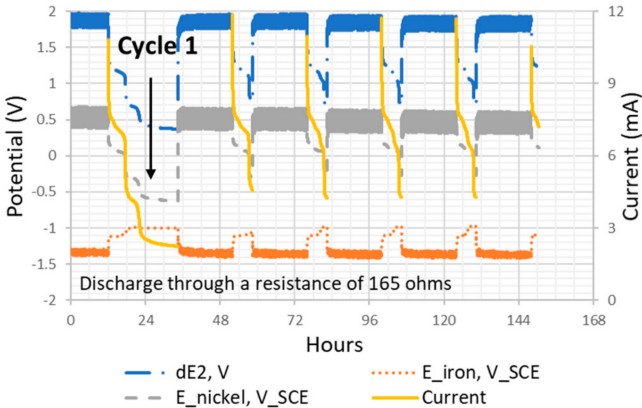

**Figure 8.** The potential and current profiles of the Ni–Fe cell with a 165 Ω resistor. The first discharge is enlarged and presented in Figure 9. Measurements include the potential difference between the cathode and anode (dE2), the potential of the iron plate as anode (E_iron, V$_{SCE}$), the potential of the Ni–CF cathode (E_nickel, V$_{SCE}$), and the current with a 165 Ω resistor. The arrow indicates the region of cycle one.

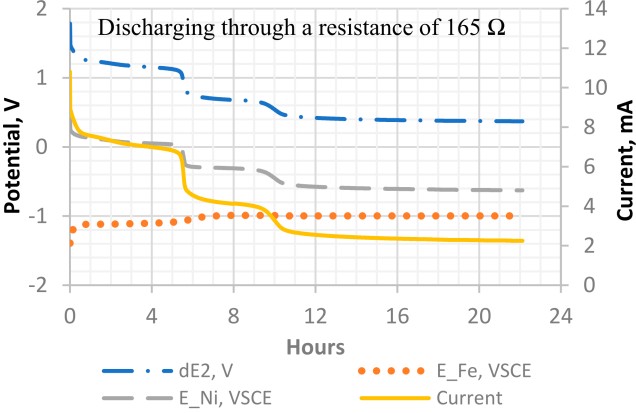

**Figure 9.** A detailed potential and current profile of discharge cycle 1 of the Ni-Fe cell. Measurements include the potential difference between the cathode and anode (dE2), the potential of the iron plate as anode (E_Fe, V$_{SCE}$), the potential of the Ni-CF cathode (E_Ni, V$_{SCE}$), and the current with a 165 Ω resistor.

Figure 9 illustrates the details from cycle one (22-h discharging) through a 165 Ω resistor. For this cycle, we prolonged the discharge period to 22 h. The results showed a very promising outcome; the current could still maintain greater than 2 mA after 22 h of discharge. The electrical capacity of discharge cycle one was 51 mAh at 8 h and 88 mAh at 22 h.

Table 12 summarizes the performance of the Ni–CF/iron plate cell and the Ni–CF/Fe–CF cell. Compared to the iron plate anode, when using the Fe–CF anode, the battery capacity was almost doubled within an even shorter discharge duration. The results confirmed that the Fe–CF anode performed better than the pure iron plate.

**Table 12.** Summary of cell performance tested in alkali solution as electrolyte.

| Cell | Discharge Cycles | Battery Capacity (mAh) | Energy (mWh) | Energy Density * (Wh/m²) |
|---|---|---|---|---|
| Ni-1/Fe-3 (Powder-mixing) | 1 (at 6 h) | 5.6 | 4.3 | 0.5 |
| Ni–CF/Fe plate | 1 (at 8 h) | 22 | 23 | 2.8 |
|  | 2 (at 8 h) | 20 | 19 | 2.3 |
| Ni–CF/Fe–CF | 1 (at 6 h) | 42 | 50 | 6.2 |
|  | 1 (at 8 h) | 51 | 56 | 6.9 |
|  | 1 (at 12) | 64 | 64 | 7.9 |
|  | 1 (at 22) | 89 | 73 | 9.0 |
|  | 2 (at 6 h) | 44 | 52 | 6.4 |
|  | 3 (at 6 h) | 43 | 49 | 6.1 |
|  | 4 (at 6 h) | 43 | 49 | 6.1 |
|  | 5 (at 6 h) | 43 | 49 | 6.1 |

* The area of the electrode is 9 cm × 9 cm.

### 4.3.3. Ni–CF Anode vs. Zn–CF Cathode

The cement-based cathode with Ni–CF mesh was tested against a cement-based anode with Zn–CF mesh. Figure 10 presents two charge and discharge cycles of the Ni–Zn cell in a no-load condition. Figure 11 demonstrates the details from the first 8-h discharge period with no resistor. The potential difference between the two electrodes (dE1) dropped down quickly shortly after the discharge cycle began, which means that it could not maintain the potential level. As the results indicated, the Zn anode was not a good candidate for the application of concrete batteries, probably due to its easy formation of zinc oxide even during the mixing with cement.

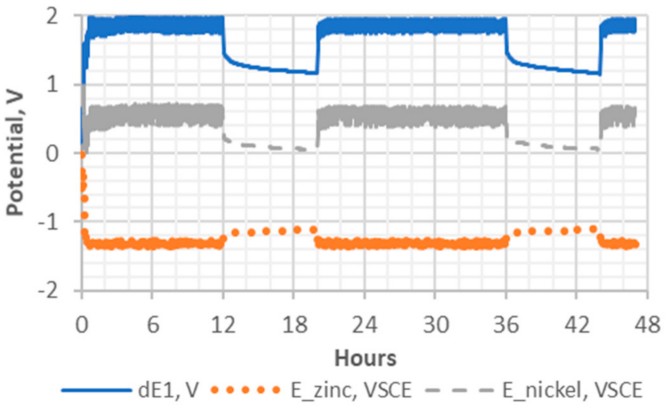

**Figure 10.** Potential profiles of the Ni-Zn cell during charge and discharge cycles under a no-load condition. Curve dE1 represents the potential difference between the cathode and the anode. The potentials of the anode and cathode are indicated by E_zinc and E_nickel, respectively.

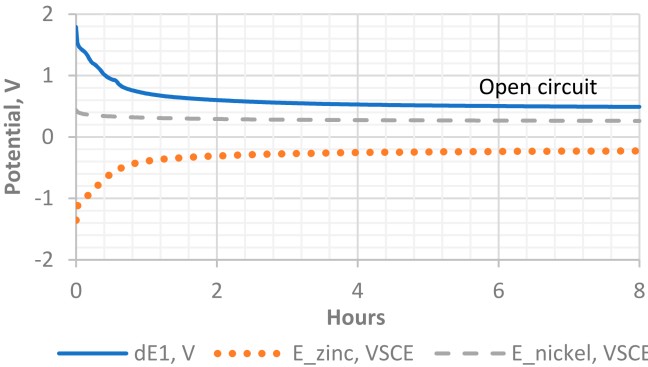

**Figure 11.** The open-circuit potential of the Ni–Zn cell during discharge. Curve dE1 represents the potential difference between the cathode and the anode. The potentials of the anode and cathode are indicated by E_zinc and E_nickel, respectively.

*4.4. Ni–Fe Concrete Battery*

A single-cell cement-based battery was build based on the best combination we gained in this work, which was an Ni–CF cathode and Fe–CF anode in conductive cementitious mortar. It is worth emphasizing that the concrete battery was tested in a hardened cement-based electrolyte separator, which was different from the cells using alkaline solution as electrolyte. The performance of the cementitious separator has been summarized in another article. Hence, we only present the performance of the three-layered concrete battery as a whole.

The discharge period was 12 h. Figure 12 presents the performance of the concrete battery under six charge and discharge cycles. The average battery output of the six cycles was 60 mAh. The detailed data are listed in Table 13. A detailed discharge profile of cycle one is given in Figure 13. The cell voltage (dE2) showed a typical plateau profile, which was representative of two-step discharging, indicating a change in the reaction mechanism and potential of the active materials [16]. Figure 14 shows an image of the real battery sample, and the cell potential was 1.24 V measured at 3-h of discharging.

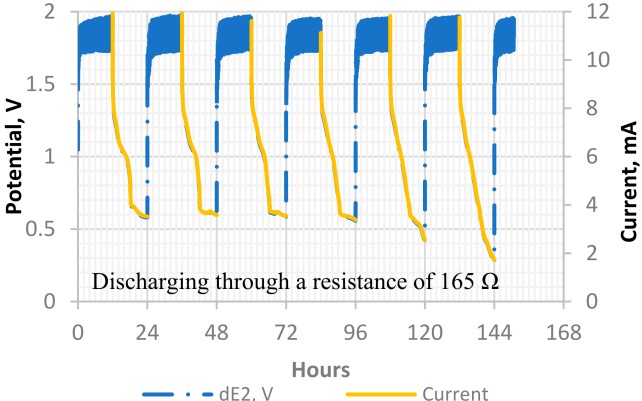

**Figure 12.** Six charge and discharge cycles of the Ni–Fe concrete battery, through a resistance of 165 Ω. The cell potential is represented by dE2, which is the potential difference between the Ni cathode and the Fe anode.

**Table 13.** Battery performance of Ni-Fe concrete battery. Each discharge cycle is 12 h.

| Discharging Cycles | Battery Capacity (mAh) | Energy (mWh) | Energy Density * (Wh/m$^2$) |
|---|---|---|---|
| 1 | 62 | 57.6 | 7.10 |
| 2 | 62 | 58.5 | 7.23 |
| 3 | 61 | 57.3 | 7.07 |
| 4 | 61 | 56.3 | 6.95 |
| 5 | 59 | 53.7 | 6.63 |
| 6 | 55 | 49.5 | 6.11 |

* The area of the electrode is 9 cm $\times$ 9 cm.

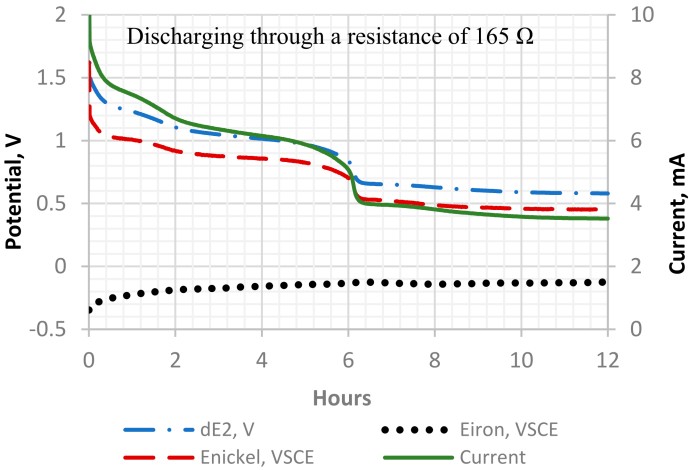

**Figure 13.** A detailed potential and current development of discharge cycle one. The cell potential is represented by dE2, which is the potential difference between the Ni cathode and the Fe anode.

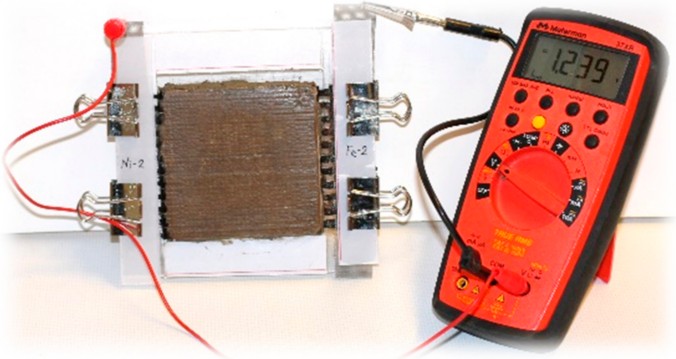

**Figure 14.** An image of the single-cell cement-based structural battery. A multimeter shows the cell potential of 1.24 V after a 3-h discharge.

## 5. Concluding Remarks

From this work, we have obtained the following results:

- The electronic resistivity in a cement-based electrolyte with the addition of short carbon fiber by 0.9 vol% and CF dispenser is about 0.9 $\Omega\cdot$m, compared to the reference of 11 $\Omega\cdot$m (7 days after mixing).
- The cement-based electrodes with iron powder and Ni (OH)$_2$ powder are rechargeable in liquid electrolytes and can make a small LED lamp shine for several hours. However, its energy density is only 0.5 Wh/m$^2$ or 0.06 Wh/L.

- The cement-based electrodes with a coating of metal Ni and Fe, respectively, by the electroplating technique can have a maximum energy density (in a liquid electrolyte) of 7 Wh/m$^2$ and an average value of 6.8 Wh/m$^2$ during six charge/discharge cycles.
- Zn was not suitable as a concrete battery anode.

Our first rechargeable cement-based battery revealed an energy density of approximately 7 Wh/m$^2$ or 0.8 Wh/L. The results clearly showed that the cement-based electrodes with electroplating of metal were much better than those mixed with metal or metal oxide powders. Although the energy density of 0.8 Wh/L was markedly lower than the commercial batteries, there is a great opportunity to build rechargeable cement-based batteries on a large scale, with regard to the huge volume of a building.

**Author Contributions:** Conceptualization, E.Q.Z. and L.T.; Formal analysis, E.Q.Z.; Funding acquisition, L.T.; Investigation, E.Q.Z. and L.T.; Methodology, E.Q.Z. and L.T.; Project administration, L.T.; Visualization, E.Q.Z.; Writing—original draft, E.Q.Z.; Writing—review and editing, E.Q.Z. and L.T. All authors have read and agreed to the published version of the manuscript.

**Funding:** This research was funded by Energimyndigheten (the Swedish Energy Agency), grant number 44748-1.

**Institutional Review Board Statement:** Not applicable.

**Informed Consent Statement:** Not applicable.

**Data Availability Statement:** The data presented in this study are available in this article.

**Acknowledgments:** This study was supported by Energimyndigheten (the Swedish Energy Agency) with project number 44748-1 and reference number 2017-005118.

**Conflicts of Interest:** The authors declare no conflict of interest. The funders had no role in the design of the study; in the collection, analyses, or interpretation of data; in the writing of the manuscript; or in the decision to publish the results.

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
