# Peer review of "Rechargeable Concrete Battery"

_buildings, doi:10.3390/buildings11030103_

Round 1

Reviewer 1 Report

The paper entitled ” Rechargeable concrete battery” deals with an important subject as it presents a possibility of using structures and buildings as energy source and storage.

The Abstract is well structured and it includes on short the aim of the paper, the study methods and its results.

Introduction presents the general notions, in order to make a general view on the subject and to understand its importance. The Introduction concludes with  the aim of the paper.

In Methodology chapter, there are described in detail the methods applied in conducting the study.

Regarding the Results chapter, I have a question however: why the authors excluded some results off the study, when it can be done a statistical approach? I know that in a study, all results must be taken into account and are excluded only those that are too far from average. The variation of the obtained results it is advisable to be expressed in percent of growth or decrease, not only ”better”. The last sentence of the first paragraph from 4.3.1 Ni-CF anode vs. Iron plate cathode, must be reexamined; it is not clear.

The Conclusions highlights the results obtained.

As supplementary mentions, there are small typing errors to be as example in first paragraph of the Introduction, the last paragraph before of subchapter 2.2, the last paragraph before of subchapter 2.3, or the reaction from table 2. In Table 3, columns or text must be resized in order that in first column to be met full words on a row. English language must be slightly rechecked.

Author Response

Thank you for your comments. Regarding the concerning in the results part, I have corrected the last sentence in section 4.3.1 Ni-CF anode vs. Iron plate cathode, it was a mistake in the figure number. Regarding the concern of excluding results, that was not the case in our paper. For example, in the section 4.2 and 4.3, the reason we wrote the sentences like "here we only present a few results as they were representative of the other results" is because the results of different cells were really similar, if we present all of them, it is only repetition. That is why we think it is enough to present one representative figures but not all of the figures.

I have also corrected all the typos in the manuscript. Thank you for pointing that out. The format of tables and figures has been readjusted according to the template.

Reviewer 2 Report

            The buildings-1125680 manuscript deals about the use of cement as solid electrolyte for Ni-Fe battery. The authors claim that the cement-based electrodes with iron powder and Ni (OH)2 powder are rechargeable in liquid electrolytes but with low energy density. Instead, the cement-based electrodes with coating of metal Ni and Fe (in a liquid electrolyte) have better performance. The manuscript is well organized to discuss the results and set conclusions. Notwithstanding, the authors should accomplish these minor revisions before the final manuscript could be accepted for the publication in Buildings.

- The authors propose two different uses for cement in rechargeable cells. Either as solid electrolyte or conductive filler in the electrode. Based on this, they propose different combinations of them and also a liquid electrolyte to prepared the test cells. Taking into account, that the conductive properties of cell electrolytes and electrodes are opposites, the reading of the results and discussion section is somehow confusing. The opinion of the reviewer is to focus on one of these uses.

- The authors should provide reference data about the electrochemical performance of related cells to be compared to the achievements of this work.

- Please check subscript in chemical formulae.

- Figures are not finely prepared. For instance, legends are really confusing.

Author Response

Thank you for your comments. Here I reply each comment after the reviewer's text.

- The authors should provide reference data about the electrochemical performance of related cells to be compared to the achievements of this work.

Answer: In the introduction, I have listed several previous study and their results by different authors. First of all, all of the previous studies were non-rechargeable cells and within my knowledge our study provided the first experimental data of rechargeable concrete battery. Secondly, there is some difficulty to compare reference data from previous studies, because of the data were represented by different unit. The conversion of different units may lead the data to error because estimation is inevitable. That is why in our study, we try to provide the data in different units, including potential, current, energy density in area, in volume. Hopefully, this will offer more reliable comparison in the future. However, I did try to convert the reference data from previous studies. With some uncertainty from the conversion, the performance of our battery should be 10 times to 100 times better than the reference data. Nonetheless, these are only estimations and not precise so that I can't state them in the paper.

- Please check subscript in chemical formulae.

Answer: the formula in table 2 was corrected and other chemical formulae were double checked too.

- Figures are not finely prepared. For instance, legends are really confusing.

Answer: All figures are clearly presented, with clear legends. Curves are presented with different line formats so even they were shown in grey scale one can still clearly differ them.

Reviewer 3 Report

Except some typos, this is a good manuscript.

The reviewer is not quite convinced about the prospect of this technology. The building is large so one can get decent energy despite low energy density. That is OK. But the cost per kWh is not mentioned. No-one will invest a large sum to get several mWh. It isn't a convincing economic case.

However, this topic of research deserves some efforts. One might find out better energy density and cost per kWh in the future.

Author Response

Thank you for your comments. The typos are all corrected.

Regarding the future application and cost, we agree there is still big gap in between. The method and results we reported here is still in lab-scale and we are aware of the future challenges for large-scale application. If we speak of the technical readiness level, this work will be standing on level 1 or 2 which is in the very beginning of the development phase. There is still a long way from commercialization and large-scale application.